# Infinite Hidden Semi-Markov Modulated Interaction Point Process

**Peng Lin**[§†]**, Bang Zhang**[§]**, Ting Guo**[§]**, Yang Wang**[§]**, Fang Chen**[§]

[§]Data61 CSIRO, Australian Technology Park, 13 Garden Street, Eveleigh NSW 2015, Australia
[†]School of Computer Science and Engineering, The University of New South Wales, Australia
{peng.lin, bang.zhang, ting.guo, yang.wang, fang.chen}@data61.csiro.au

## Abstract

The correlation between events is ubiquitous and important for temporal events modelling. In many cases, the correlation exists between not only events' emitted observations, but also their arrival times. State space models (e.g., hidden Markov model) and stochastic interaction point process models (e.g., Hawkes process) have been studied extensively yet separately for the two types of correlations in the past. In this paper, we propose a Bayesian nonparametric approach that considers both types of correlations via unifying and generalizing the hidden semi-Markov model and interaction point process model. The proposed approach can simultaneously model both the observations and arrival times of temporal events, and automatically determine the number of latent states from data. A Metropolis-within-particle-Gibbs sampler with ancestor resampling is developed for efficient posterior inference. The approach is tested on both synthetic and real-world data with promising outcomes.

## 1 Introduction

Temporal events modeling is a classic machine learning problem that has drawn enormous research attentions for decades. It has wide applications in many areas, such as financial modelling, social events analysis, seismological and epidemiological forecasting. An event is often associated with an arrival time and an observation, e.g., a scalar or vector. For example, a trading event in financial market has a trading time and a trading price. A message in social network has a posting time and a sequence of words. A main task of temporal events modelling is to capture the underlying events correlation and use it to make predictions for future events' observations and/or arrival times.

The correlation between events' observations can be readily found in many real-world cases in which an event's observation is influenced by its predecessors' observations. For examples, the price of a trading event is impacted by former trading prices. The content of a new social message is affected by the contents of the previous messages. State space model (SSM), e.g., the hidden Markov model (HMM) [16], is one of the most prevalent frameworks that consider such correlation. It models the correlation via latent state dependency. Each event in the HMM is associated with a latent state that can emit an observation. A latent state is independent of all but the most recent state, *i.e.*, Markovianity. Hence, a future event observation can be predicted based on the observed events and inferred mechanisms of emission and transition.

Despite its popularity, the HMM lacks the flexibility to model event arrival time. It only allows fixed inter-arrival time. The duration of a type of state follows a geometric distribution with its self-transition probability as the parameter due to the strict Markovian constraint. The hidden semi-Markov model (HSMM) [14, 21] was developed to allow non-geometric state duration. It is an extension of the HMM by allowing the underlying state transition process to be a semi-Markov chain

with a variable duration time for each state. In addition to the HMM components, the HSMM models the duration of a state as a random variable and a state can emit a sequence of observations.

The HSMM allows the flexibility of variable inter-arrival times, but it does not consider events' correlation on arrival times. In many real-world applications, one event can trigger the occurrences of others in the near future. For instance, earthquakes and epidemics are diffusible events, *i.e.*, one can cause the occurrences of others. Trading events in financial markets arrive in clusters. Information propagation in social network shows contagious and clustering characteristics. All these events exhibit interaction characteristics in terms of arrival times. The likelihood of an event's arrival time is affected by the previous events' arrival times. Stochastic interaction point process (IPP), *e.g.*, Hawkes process [6], is a widely adopted framework for capturing such arrival time correlation. It models the correlation via a conditional intensity function that depicts the event intensity depending on all the previous events' arrival times. However, unlike the SSMs, it lacks the capability of modelling events' latent states and their interactions.

It is clearly desirable in real-world applications to have both arrival time correlation and observation correlation considered in a unified manner so that we can estimate both when and how events will appear. Inspired by the merits of SSMs and IPPs, we propose a novel Bayesian nonparametric approach that unifies and generalizes SSMs and IPPs via a latent semi-Markov state chain with infinitely countable number of states. The latent states governs both the observation emission and new event triggering mechanism. An efficient sampling method is developed within the framework of particle Markov chain Monte Carlo (PMCMC) [1] for the posterior inference of the proposed model.

We first review closely related techniques in Section 2, and give the description of the proposed model in Section 3. Then Section 4 presents the inference algorithm. In Section 5, we show the results of the empirical studies on both synthetic and real-word data. Conclusions are drawn in Section 6.

# 2 Preliminaries

In this section, we review the techniques that are closely related to the proposed method, namely hidden (semi-)Markov model, its Bayesian nonparametric extension and Hawkes process.

## 2.1 Hidden (Semi-)Markov Model

The HMM [16] is one of the most popular approaches for temporal event modelling. It utilizes a sequence of latent states with Markovian property to model the dynamics of temporal events. Each event in the HMM is associated with a latent state that determines the event's observation via a emission probability distribution. The state of an event is independent of all but its most recent predecessor's state (*i.e.*, Markovianity) following a transition probability distribution. The HMM consists of 4 components: (1) an initial state probability distribution,(2) a finite latent state space, (3) a state transition matrix, and (4) an emission probability distribution. As a result, the inference for the HMM involves: inferring (1) the initial state probability distribution, (2) the sequence of the latent states, (3) the state transition matrix and (4) the emission probability distribution.

The HMM has proven to be an excellent general framework modelling sequential data, but it has two significant drawbacks: (1) The durations of events (or the inter-arrival times between events) are fixed to a common value. The state duration distributions are restricted to a geometric form. Such setting lacks the flexibility for real-world applications. (2) The size of the latent state space in the HMM must be set a priori instead of learning from data.

The hidden semi-Markov model (HSMM) [14, 21] is a popular extension to the HMM, which tries to mitigate the first drawback of the HMM. It allows latent states to have variable durations, thereby forming a semi-Markov chain. It reduces to the HMM when durations follow a geometric distribution. Additional to the 4 components of the HMM, HSMM has a state duration probability distribution. As a result, the inference procedure for the HSMM also involves the inference of the duration probability distribution. It is worth noting that the interaction between events in terms of event arrival time is neglected by both the HMM and the HSMM.

## 2.2 Hierarchical Dirichlet Process Prior for State Transition

The recent development in Bayesian nonparametrics helps address the second drawback of the HMM. Here, we briefly review the Hierarchical Dirichlet Process HMM (HDP-HMM). Let $(\Theta, \mathcal{B})$ be a measurable space and $G_0$ be a probability measure on it. A Dirichlet process (DP) $G$ is a distribution of a random probability measure over the measurable space $(\Theta, \mathcal{B})$. For any finite measurable partition $(A_1, \cdots, A_r)$ of $\Theta$, the random vector $(G(A_1), \cdots, G(A_r))$ follows a finite Dirichlet distribution parameterized by $(\alpha_0 G_0(A_1), \cdots, \alpha_0 G_0(A_r))$, where $\alpha_0$ is a positive real number.

HDP is defined based on DP for modelling grouped data. It is a distribution over a collection of random probability measures over the measurable space $(\Theta, \mathcal{B})$. Each one of these random probability measure $G_k$ is associated with a group. A global random probability measure $G_0$ distributed as a DP is used as a mean measure with concentration parameter $\gamma$ and base probability measure $H$. Because the HMM can be treated as a set of mixture models in a dynamic manner, each of which corresponds to a value of the current state, the HDP becomes a natural choice as the prior over the state transitions [2, 18]. The generative HDP-HMM model can be summarized as:

$$
\begin{aligned}
\beta \mid \gamma &\backsim \mathrm{GEM}(\gamma), \quad \pi_k \mid \alpha_0, \beta \backsim \mathrm{DP}(\alpha_0, \beta), \quad \theta_k \mid \lambda, H \backsim H(\lambda), \\
s_n \mid s_{n-1}, (\pi_k)_{k=1}^{\infty} &\backsim \pi_{s_{n-1}}, \quad y_n \mid s_n, (\theta_k)_{k=1}^{\infty} \backsim F(\theta_{s_n})
\end{aligned}
\tag{1}
$$

GEM denotes stick-breaking process. The variable sequence $\pi_k$ indicates the latent state sequence. $y_n$ represents the observation. HDP acts the role of a prior over the infinite transition matrices. Each $\pi_k$ is a draw from a DP, it depicts the transition distribution from state $k$. The probability measures from which $\pi_k$'s are drawn are parameterized by the same discrete base measure $\beta$. $\theta$ parameterizes the emission distribution $F$. Usually $H$ is set to be conjugate of $F$ simplifying inference. $\gamma$ controls base measure $\beta$'s degree of concentration. $\alpha_0$ plays the role of governing the variability of the prior mean measure across the rows of the transition matrix.

Because the HDP prior doesn't distinguish self-transitions from transitions to other states, it is vulnerable to unnecessary frequent switching of states and more states. Thus, [5] proposed a sticky HDP-HMM to include a self-transition bias parameter into the state transition measure $\pi_k \sim DP(\alpha_0 + \kappa, (\alpha_0 \beta + \kappa \delta_k)/(\alpha_0 + \kappa))$, where $\kappa$ controls the stickness of the transition matrix.

## 2.3 Hawkes Process

Stochastic point process [3] is a rich family of models that are designed for tackling various of temporal event modeling problems. A stochastic point process can be defined via its conditional intensity function that provides an equivalent representation as a counting process for temporal events. Given $N(t)$ denoting the number of events occurred in the time interval $[0, t)$ and $\tau_t$ indicating the arrival times of the temporal events before $t$, the intensity for a time point $t$ conditioned on the arrival times of all the previous events is defined as:

$$
\lambda(t \mid \tau_t) = \lim_{\Delta t \to 0} \frac{\mathbb{E}[N(t + \Delta t) - N(t) \mid \tau_t]}{\Delta t}.
\tag{2}
$$

It is worth noting that we do not consider edge effect in this paper, hence no events exist before time 0. A variety of point processes has been developed with distinct functional forms of intensity for various modeling purposes. Interaction point process (IPP) [4] considers point interactions with an intensity function dependent on historical events. Hawkes process [7, 6] is one of the most popular and flexible IPPs. Its conditional intensity has the following functional form:

$$
\lambda(t) = \mu(t) + \sum_{t_n < t} \psi_n(t - t_n).
\tag{3}
$$

We use $\lambda(t)$ to represent intensity function conditioned on previous points $\tau_t$ with the consideration of notation simplicity. The function $\mu(t)$ is a non-negative background intensity function which is often set to a positive real number. The function $\psi_n(\triangle t)$ represents the triggering kernel of event $t_n$. It is a decay function defined on $[0, \infty)$ depicting the decayed influence of triggering new events. A typical decay function is in exponential form, *i.e.*, $\lambda(t) = \mu + \sum_{t_n < t} \alpha' \cdot \exp(-\beta'(t - t_n))$. As discussed in [7, 10], because the superposition of several Poisson processes is also a Poisson process, Hawkes process can be considered as a conditional Poisson process that is a constituted by combining a background Poisson process $\mu(t)$ and a set of triggered Poisson processes with intensity $\psi_n(t - t_n)$.

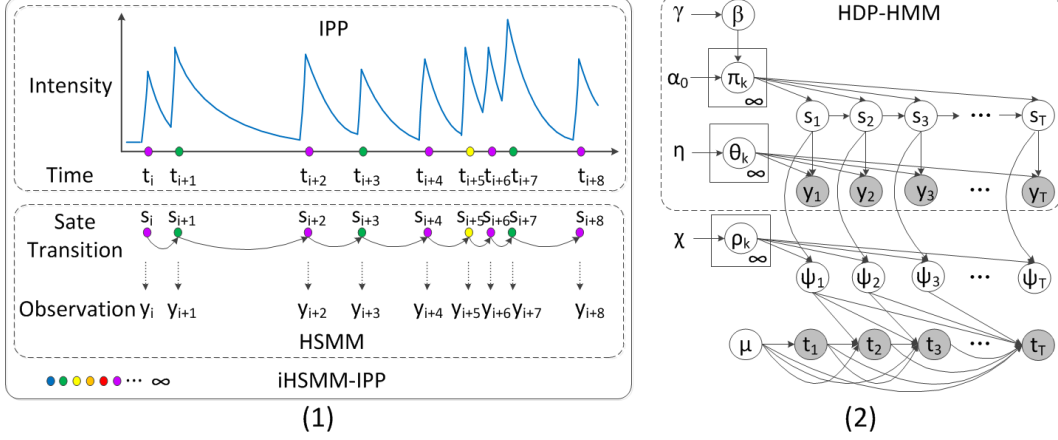

Figure 1: (1) An intuitive illustration of the iHSMM-IPP model. Every event in the iHSMM-IPP model is associated with a latent state $s$, an arrival time $t$ and an observable value $y$. The colours of points indicate latent states. Blue curve shows the event intensity. The top part of the figure illustrates the IPP component of the iHSMM-IPP model and the bottom part illustrates the HSMM component. The two components are integrated together via an infinite countable semi-Markov latent state chain. (2) Graphical model of the iHSMM-IPP model. The top part shows the HDP-HMM.

## 3 Infinite Hidden Semi-Markov Modulated Interaction Point Process (iHSMM-IPP)

Inspired by the merits of SSMs and IPPs, we propose an infinite hidden semi-Markov modulated interaction point process model (iHSMM-IPP). It is a Bayesian nonparametric stochastic point process with a latent semi-Markov state chain determining both event emission probabilities and event triggering kernels. An intuitive illustration is given in Fig. 1 (1). Each temporal event in the iHSMM-IPP is represented by a stochastic point and each point is associated with a hidden discrete state $\{s_i\}$ that plays the role of determining event emission and triggering mechanism. As in SSMs and IPPs, the event emission probabilities guide the generation of event observations $\{y_i\}$ and the event triggering kernels influence the occurrence times $\{t_i\}$ of events. The hidden state depends only on the most recent event's state. The size of the latent state space is infinite countable with the HDP prior.

The model can be formally defined as the following and its corresponding graphical model is given in Fig. 1 (2).

$$
\begin{aligned}
&\beta \mid \gamma \backsim \text{GEM}(\gamma), \quad \pi_k \mid \alpha_0, \beta \backsim \text{DP}(\alpha_0, \beta), \quad \theta_k \mid \eta, H \backsim H(\eta), \\
&\qquad \rho_k \mid \chi, H' \backsim H'(\chi), \quad s_n \mid s_{n-1}, (\pi_k)_{k=1}^{\infty} \backsim \pi_{s_{n-1}}, \\
&t_n \mid \cdot \backsim \mathcal{PP}(\mu + \sum_{i=1}^{n-1} \psi_{\rho_{s_i}}(t - t_i)), \quad y_n \mid s_n, (\theta_k)_{k=1}^{\infty} \backsim F(\theta_{s_n}).
\end{aligned}
\tag{4}
$$

We use $\psi_{\rho_{s_i}}(\cdot)$ to denote the triggering kernel parameterized by $\rho_{s_i}$ which is indexed by latent state $s_i$. We use $\psi_{s_i}(\cdot)$ instead of $\psi_{\rho_{s_i}}(\cdot)$ for the remaining of the paper for the sake of notation simplicity. The iHSMM-IPP is a generative model that can be used for generating a series of events with arrival times and emitted observations. The arrival time $t_n$ is drawn from a Poisson process. We do not consider edge effect in this work. Therefore, the first event's arrival time, $t_1$, is drawn from a homogeneous Poisson process parameterized by a hyper-parameter $\mu$. For $n > 1$, $t_n$ is drawn from an inhomogeneous Poisson process whose conditional intensity function is defined as: $\mu + \sum_{i=1}^{n-1} \psi_{s_i}(t - t_i)$. As defined before, $\psi_{s_i}(\cdot)$ indicates the triggering kernel of a former point $i$ whose latent state is $s_i$. The state of the point $s_n$ is drawn following the guidance of the HDP prior as in the HDP-HMM. The emitted observation $y_n$ is generation from the emission probability distribution $F(\cdot)$ parameterized by $\theta_{s_n}$ which is determined by the state $s_n$.

# 4 Posterior Inference for the iHSMM-IPP

In this section, we describe the inference method for the proposed iHSMM-IPP model. Despite its flexibility, the proposed iHSMM-IPP model faces three challenges for efficient posterior inference: (1) strong correlation nature of its temporal dynamics (2) non-Markovianity introduced by the event triggering mechanism, and (3) infinite dimensional state transition. The traditional sampling methods for high dimensional probability distributions, *e.g.*, MCMC, sequential Monte Carlo (SMC), are unreliable when highly correlated variables are updated independently, which can be the case for the iHSMM-IPP model. So we develop the inference algorithm within the framework of particle MCMC (PMCMC), a family of inferential methods recently developed in [1]. The key idea of PMCMC is to use SMC to construct a proposal kernel for an MCMC sampler. It not only improves over traditional MCMC methods but also makes Bayesian inference feasible for a large class of statistical models. For tackling the non-Markovianity, ancestor resampling scheme [13] is incorporated into our inference algorithm. As existing forward-backward sampling methods, ancestor resampling uses backward sampling to improve the mixing of PMCMC. However, it achieves the same effect in a single forward sweep instead of using separate forward and backward sweeps. More importantly, it provides an effective way of sampling for non-Markovian SSMs.

Given a sequence of $N$ events, $\{y_n, t_n\}_{n=1}^N$, the inference algorithm needs to sample the hidden state sequence, $\{s_n\}_{n=1}^N$, emission distribution parameters $\theta_{1:K}$, background event intensity $\mu$, triggering kernel parameters, $\psi_{1:K}$ (we omit $\rho$ and use $\psi_{1:K}$ instead of $\psi_{\rho_{1:K}}$ for notation simplicity as before), transition matrix, $\boldsymbol{\pi}_{1:K}$, and the HDP parameters $(\alpha_0, \gamma, \kappa, \beta)$. We use $K$ to represent the number of active states and $\Omega$ to indicate the set of variables excluding the latent state sequence, *i.e.*, $\Omega = \{\alpha_0, \beta, \gamma, \kappa, \mu, \theta_{1:K}, \psi_{1:K}, \pi_{1:K}\}$. Only major variables are listed, and $\Omega$ may also include other variables, such as the probability of initial latent state. At a high level, all the variables are updated iteratively using a particle Gibbs (PG) sampler. A conditional SMC is performed as a proposal kernel for updating latent state sequence in each PG iteration. An ancestor resampling scheme is adopted in the conditional SMC for handling the non-Markovianity caused by the triggering mechanism. Metropolis sampling is used in each PG iteration to update background event intensity $\mu$ and triggering kernel parameters $\psi_{1:K}$. The remaining variables in $\Omega$ can be sampled by following the scheme in [5, 18] readily. The proposal distribution $q_\Omega(\cdot)$ in the conditional SMC can be set by following [19]. The PG sampler is given in the following:

**Step 1:** Initialization, $i = 0$, set $\Omega(0)$, $s_{1:N}(0)$, $B_{1:N}(0)$.

**Step 2:** For iteration $i \geqslant 1$

    **(a)** Sample $\Omega(i) \sim p\{\cdot | y_{1:N}, t_{1:N}, s_{1:N}(i-1)\}$.

    **(b)** Run a conditional SMC algorithm targeting $p_{\Omega(i)}(s_{1:N} | y_{1:N}, t_{1:N})$ conditional on $s_{1:N}(i-1)$ and $B_{1:N}(i-1)$.

    **(c)** Sample $s_{1:N}(i) \sim \hat{p}_{\Omega(i)}(\cdot | y_{1:N}, t_{1:N})$.

We use $B_{1:N}$ to represent the ancestral lineage of the prespecified state path $s_{1:N}$ and $\hat{p}_{\Omega(i)}(\cdot | y_{1:N})$ to represent the particle approximation of $p_{\Omega(i)}(\cdot | y_{1:N})$. The details of the conditional SMC algorithm are given in the following. It is worth noting that the conditioned latent state path is only updated via the ancestor resampling.

**Step 1:** Let $s_{1:N} = \{s_1^{B_1}, s_2^{B_2}, \cdots, s_N^{B_N}\}$ denote the path that is associated with the ancestral lineage $B_{1:N}$

**Step 2:** For $n = 1$,

    **(a)** For $j \neq B_1$, sample $s_1^j \sim q_\Omega(\cdot | y_1)$, $j \in [1, \cdots, J]$. ($J$ denotes the number of particles.)

    **(b)** Compute weights $w_1(s_1^j) = p(s_1^j) F(y_1 | s_1^j) / q_\Omega(s_1^j | y_1)$ and normalize the weights $W_1^j = w_1(s_1^j) / \sum_{m=1}^J w_1(s_1^m)$. (We use $p(s_1^j)$ to represent the probability of the initial latent state and $q_\Omega(s_1^j | y_1)$ to represent the proposal distribution conditional on the variable set $\Omega$.)

**Step 3:** For $n = 2, \cdots, N$:

    **(a)** For $j \neq B_n$, sample ancestor index of particle $j$: $a_{n-1}^j \sim \mathcal{C}at(\cdot | W_{n-1}^{1:J})$.

**(b)** For $j \neq B_n$, sample $s_n^j \sim q_\Omega(\cdot|y_n, s_{n-1}^{a_{n-1}^j})$. If $s_n^j = K+1$ then create a new state using the stick-breaking construction for the HDP:

    **(i)** Sample a new transition probability $\pi_{K+1} \sim \mathcal{D}ir(\alpha_0 \beta)$.

    **(ii)** Use stick-breaking construction to expand $\beta \leftarrow [\beta, \beta_{K+1}]$:

$$\beta'_{K+1} \sim \text{Beta}(1, \gamma), \quad \beta_{K+1} = \beta'_{K+1} \prod_{l=1}^{K}(1 - \beta'_l).$$

    **(iii)** Expand transition probability vectors $\pi_k$ to include transitions to state $K+1$ via the HDP stick-breaking construction:

$$\pi_k \leftarrow [\pi_{k,1}, \cdots, \pi_{k,K+1}], \ \ \forall k \in [1, K+1], \text{where}$$

$$\pi'_{k,K+1} \sim \text{Beta}(\alpha_0 \beta_{K+1}, \alpha_0 (1 - \sum_{l=1}^{K+1} \beta_l)), \quad \pi_{k,K+1} = \pi'_{k,K+1} \prod_{l=1}^{K}(1 - \pi'_{k,l}).$$

    **(iv)** Sample parameters for a new emission probability and triggering kernel $\theta_{K+1} \sim H$ and $\psi_{1:K} \sim H'$.

**(d)** Perform ancestor resampling for the conditioned state path. Compute the ancestor weights $\tilde{w}_{n-1|N}^{p,j}$ via Eq. 7 and Eq. 8 and resample $a_n^{B_n}$ as $p(a_n^{B_n} = j) \propto \tilde{w}_{n-1|N}^{p,j}$.

**(e)** Compute and normalize particle weights:

$$w_n(s_n^j) = \pi(s_n^j|s_{n-1}^{a_{n-1}^j})F(y_n|s_n^j)/q_\Omega(s_n^j|s_{n-1}^{a_{n-1}^j}, y_n), \ W_n(s_n^j) = w_n(s_n^j)/(\sum_{j=1}^{J} w_n(s_n^j)).$$

## 4.1 Metropolis Sampling for Background Intensity and Triggering Kernel

For the inference of the background intensity $\mu$ and the parameters of triggering kernels $\psi_k$ in the step 2 (a) of the PG sampler, Metropolis sampling is used. As described in [3], the conditional likelihood of the occurrences of a sequence of events in IPP can be expressed as:

$$\mathcal{L} \triangleq p(t_{1:N}|\mu, \psi_{1:K}) = \left(\prod_{n=1}^{N} \lambda(t_n)\right) \exp\left(-\int_0^T \lambda(t)dt\right). \tag{5}$$

We describe the Metropolis update for $\psi_k$, and similar update can be derived for $\mu$. The normal distribution with the current value of $\psi_k$ as mean is used as the proposal distribution. The proposed candidate $\psi_k^*$ will be accepted with the probability: $A(\psi_k^*, \psi_k) = \min\left(1, \frac{\hat{p}(\psi_k^*)}{\hat{p}(\psi_k)}\right)$. The ratio can be computed as:

$$\frac{\hat{p}(\psi_k^*)}{\hat{p}(\psi_k)} = \frac{p(\psi_k^*)}{p(\psi_k)} \cdot \frac{p(t_{1:N}|\psi_k^*, \text{rest})}{p(t_{1:N}|\psi_k, \text{rest})} = \frac{p(\psi_k^*)}{p(\psi_k)} \cdot \left(\prod_{n=1}^{N} \frac{\mu(t_n) + \sum_{u<n} \psi_{s_u}^*(t_n - t_u)}{\mu(t_n) + \sum_{u<n} \psi_{s_u}(t_n - t_u)}\right)$$

$$\cdot \exp\left(\sum_{u \in [1,N]} (\Psi_{s_u}(T - t_u) - \Psi_{s_u}^*(T - t_u))\right). \tag{6}$$

We use $\Psi(\cdot)$ to represent the cumulative distribution function of the kernel function $\psi(\cdot)$. We use $\psi_{s_u}^*(\cdot)$ to represent the $u$-th event's triggering kernel candidate if $s_u = k$. It remains the current triggering kernel otherwise. $[0, T]$ indicates the time period of the $N$ events.

## 4.2 Truncated Ancestor Resampling for Non-Markovianity

Truncated ancestor resampling [13] is used for tackling the non-Markovianity caused by the triggering mechanism of the proposed model. The ancestor weight can be computed as:

$$\tilde{w}_{n|N}^{p,j} = w_n^j \frac{\gamma_{n+p}(\{s_{1:n}^j, s'_{n+1:n+p}\})}{\gamma_n(s_{1:n}^j)} \tag{7}$$

$$\frac{\gamma_{n+p}(\{s_{1:n}^j, s'_{n+1:n+p}\})}{\gamma_t(s_{1:n}^j)} = \frac{p(s_{1:p}, y_{1:p}, t_{1:p})}{p(s_{1:n}, y_{1:n}, t_{1:n})} = \frac{\mathcal{L}(t_{1:p})}{\mathcal{L}(t_{1:n})} \cdot \prod_{j=n+1}^{p} F(y_j|s_j)\pi(s_j|s_{j-1}) \tag{8}$$

For notation simplicity, we use $w_n^j$ to represent $w_n(s_n^j)$. In general, $n + p$ needs to reach the last event in the sequence. However, due the computational cost and the influence decay of the past events in the proposed iHSMM-IPP, it is practical and feasible to use only a small number of events as an approximation instead of using all the remaining events in Eq. 8.

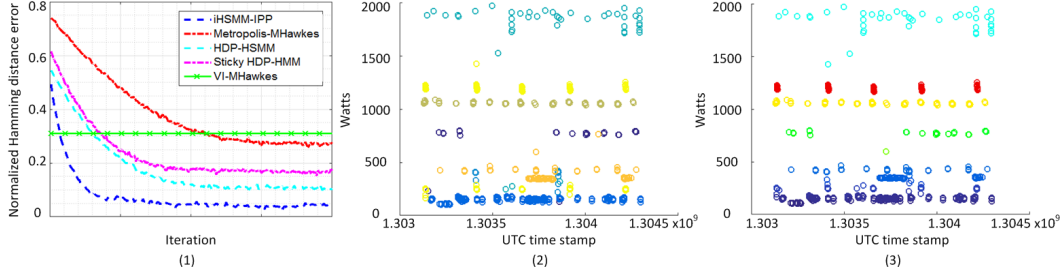

Figure 2: (1) Normalized Hamming distance errors for synthetic data. (2) Cleaned energy consumption readings of the REDD data set. (3) Estimated states by the proposed iHSMM-IPP model.

## 5 Empirical Study

In the following experiments, we demonstrate the performance of the proposed inference algorithm and show the applications of the proposed iHSMM-IPP model in real-world settings.

### 5.1 Synthetic Data

As in [20, 5, 19], we generate the synthetic data of $1000$ events via a 4-state Gaussian emission HMM with self-transition probability of $0.75$ and the remaining probability mass uniformly distributed over the other 3 states. The means of emission are set to $-2.0\ -0.5\ 1.0\ 4.0$ with the deviation of $0.5$. The occurrence times of events are generated via the Hawkes process with 4 different triggering kernels, each of which corresponds to a HMM state. The background intensity is set to $0.6$ and the triggering kernels take the exponential form: $\lambda(t) = 0.6 + \sum_{t_n < t} \alpha' \cdot \exp(-\beta'(t - t_n))$ with $\{0.1, 0.9\}, \{0.5, 0.9\}, \{0.1, 0.6\}, \{0.5, 0.6\}$ as the $\{\alpha', \beta'\}$ parameter pairs of the kernels. A thinning process [15] (a point process variant of rejection sampling) is used to generate event times of Hawkes process.

We compared 4 related methods to demonstrate the performance of the proposed iHSMM-IPP model and inference algorithm: particle Gibbs sampler for sticky HDP-HMM [19], weak-limit sampler for HDP-HSMM [8], Metropolis-within-Gibbs sampler for marked Hawkes process [17] and variational inference for marked Hawkes process [11]. The normalized Hamming distance error is used to measure the performance of the estimated state sequences. The Diff distance used in [22] (i.e., $\frac{\int (\tilde{\psi}(t) - \psi(t))^2 dt}{\int (\psi(t))^2 dt}$, $\psi(t)$ and $\tilde{\psi}(t)$ represent the true and estimated kernels respectively) is adopted for measuring the performance of the estimated triggering kernels. The estimated ones are greedily matched to minimize their distances from the ground truth.

The average results of the normalized Hamming distance errors are shown in Fig. 2 (1) and the Diff distance errors are shown in the second column of Table 1. The results show that the proposed inference method can not only quickly converge to an accurate estimation of the latent state sequence but also well recover the underlying triggering kernels. Its clear advantage over the compared SSMs and marked Hawkes processes is due to its considerations of both occurrence times and emitted observations for the inference.

### 5.2 Understanding Energy Consumption Behaviours of Households

In this section, we use energy consumption data from the Reference Energy Disaggregation Dataset (REDD ) [9] to demonstrate the application of the proposed model. The data set was collected via smart meters recording detailed appliance-level electricity consumption information for individual house. The data sets were collected with the intension to understand household energy usage patterns and make recommendations for efficient consumption. The 1 Hz low frequency REDD data is used and down sampled to 1 reading per minute covering 1 day energy consumption. Very low and high consumption readings are removed from the reading sequence. Fig. 2 (2) shows the cleaned reading sequence. Colours indicate appliance types and readings are in Watts.

The appliance types are modelled as latent states in the proposed iHSMM-IPP model. The readings are the emitted observations of states governed by Gaussian distributions. The relationship between the usages of different appliances is modelled via the state transition matrix. The triggering kernels

|        | Synthetic | REDD    |          | Pipe    |         |              |           |
|--------|-----------|---------|----------|---------|---------|--------------|-----------|
| Method | Diff      | Hamming | LogLik   | Hamming | LogLik  | MSE Failures | MSE Hours |
| iHSMM-IPP  | 0.36  | 0.30    | $-120.11$ | 0.39   | $-677$  | 82.8         | 28.6      |
| M-MHawkes  | 0.55  | 0.63    | $-173.36$ | 0.64   | $-1035$ | 142.2        | 80.2      |
| VI-MHawkes | 0.62  | 0.76    | $-193.62$ | 0.78   | $-1200$ | 166.7        | 93.7      |
| HDP-HSMM   | -     | 0.42    | $-147.52$ | 0.52   | $-850$  | 103.8        | 42.3      |
| S-HDP-HMM  | -     | 0.55    | $-163.28$ | 0.59   | $-993$  | 128.5        | 55.9      |

Table 1: Results on Synthetic, REDD and Pipe data sets.

of states in the model depict the influences of appliances on triggering the following energy consumptions, e.g., the usage of washing machine triggers the following energy usage of dryer. As in the first experiment, exponential form of trigger is adopted and independent exponential priors with hyper-parameter $0.01$ are used for kernel parameters $(\alpha', \beta')$.

The $4$ methods used in the first experiment are compared with the proposed model. The average results of the normalized Hamming distance errors and the log likelihoods are shown in the third and fourth columns of Table 1. The proposed model outperforms the other methods due to the fact that it has the flexibility to capture the interaction between the usages of different appliances. Other models mainly rely on the emitted observations, *i.e.*, readings for inferring the types of appliances.

### 5.3 Understanding Infrastructure Failure Behaviours and Impacts

Drinking water pipe networks are valuable infrastructure assets. Their failures (e.g., pipe bursts and leaks) can cause tremendous social and economic costs. Hence, it is of significant importance to understand the behaviours of pipe failures (i.e., occurrence time, failure type, labour hours for repair). In particular, the relationship between the types of two consecutive failures, the triggering effect of a failure on the intensity of future failures and the labour hours taken for a certain type of failure can help provide not only insights but also guidance to make informed maintenance strategies.

In this experiment, a sequence of $1600$ failures occurred in the same zone within $15$ years with $10$ different failure types [12] are used for testing the performance of the proposed iHSMM-IPP model. Failure types are modelled as latent states. Labour hours for repair are emissions of states, which are modelled by Gaussian distributions. It is well observed in industry that pipe failures occur in clusters, i.e., certain types of failures can cause high failure risk in near future. Such behaviours are modelled via the triggering kernels of states.

As in the first experiment, we compare the proposed iHSMM-IPP model with $4$ related methods. The sequence is divided into two parts $90\%$ and $10\%$. The first part of the sequence is used for training models. The normalized Hamming distance errors and log likelihoods are used for measuring the performances on the first part. Then the models are used for predicting the remaining $10\%$ of the sequence. The predicted total number of failures and total labour hours for each failure type are compared with ground truth by using mean square error. The results are shown in the last four columns of Table 1. It can be seen that the proposed iHSMM-IPP achieves the best performance for both the estimation on the first part of the sequence and the prediction on the second part of the sequence. Its superiority comes from the fact that it well utilizes both the observed labour hours and occurrence times while others only consider part of the observed information or have limitations on model flexibility.

## 6  Conclusion

In this work, we proposed a new Bayesian nonparametric stochastic point process model, namely the infinite hidden semi-Markov modulated interaction point process model. It captures both emitted observations and arrival times of temporal events for capturing the underlying event correlation. A Metropolis-within-particle-Gibbs sampler with truncated ancestor resampling is developed for the posterior inference of the proposed model. The effectiveness of the sampler is shown on a synthetic data set with the comparison of $4$ related state-of-the-art methods. The superiority of the proposed model over the compared methods is also demonstrated in two real-world applications, *i.e.*, household energy consumption modelling and infrastructure failure modelling.

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
