[Reviews · NeurIPS 2016]

Reviewer 1

Summary

In this paper, the authors combine ideas from the HDP-HSMM and temporal proint processes (in Particular, Hawkes processes) to derive an infinite HSMM in which the time between state transitions is driven by a Hawkes process. In order to infer the model, the authors propose a Gibb sampler with ancestor sampling algorithm in which some of the hyperparameters are updated using Metropolis Hastings. Finally, the authors apply the model to synthetic data and two real applications (power disaggregation and water pipe failures).

Qualitative Assessment

The proposed model is novel and has several potential applications. Additionally, the paper is well motivated and written. However, my main concerns are related to the reproducibility of the proposed inference and experiments. The authors do not provide enough details in the paper (or the non-existent supplementary material) to implement the proposed inference algorithm or reproduce their experiments. Detailed comments: 1) Inference. - As I commented above, more details on the inference are necessary to both check its correctness or implement it. - The authors should introduce the function \gamma_n() in equations (7) and (8). - I think that the impact of the paper would increase if the authors released the code. 2) Experiments: - Further explanation on the setup for the two real applications is needed. For example, how does observation model look? - In the case of power disaggregation: What are exactly the observations? What is in this case an event? a jump in the total power consumption? What are the event times modeling? The time at which a device turns on? Are the results measured over all the houses in the REDD? How is the Hamming distance computed? How are you computing the log likelihood for each model? I am not sure whether the log-likelihood is a fair measure for model checking in this case, since some of the models include the event times as observations (HSMM models) while other only accounts for the power measurements (standard HMMs). This would explain the large differences in log-likelihood. In the related work, authors have measured accuracy of the estimated total power consumption per device (see, e.g., https://arxiv.org/pdf/1203.1365.pdf, http://ieeexplore.ieee.org/xpl/articleDetails.jsp?reload=true&arnumber=7322279). Does the model estimate correctly the true number of devices and which ones are active at any given time? Moreover, I am not convinced that this is a suitable application for the proposed model. How does the model handle the fact that several devices can be working at the same time (which according to Figure 2 is very common). Several related work has proposed factorial HMMs to capture this fact (see, e.g., http://ieeexplore.ieee.org/xpl/articleDetails.jsp?reload=true&arnumber=7322279 and https://papers.nips.cc/paper/5667-infinite-factorial-dynamical-model.pdf) - In the case of failure behavior: Although I find this experimental set up for the proposed model, I think that the authors should provide further details on how the different measures for model comparisons are computed. This would allow the reader to better understand what these values mean. 3) Further related work. Maybe the authors would be interested in having a look to a related paper that combines DPs with Hawkes process: http://dl.acm.org/citation.cfm?id=2783411 3) Format. I am not sure that the submitted draft satisfies the format guidelines provided by NIPS. In particular, I think that the fontsize is smaller that the minimum permitted.

Confidence in this Review

3-Expert (read the paper in detail, know the area, quite certain of my opinion)


Reviewer 2

Summary

In this paper the authors propose a Bayesian non parametric approach to model temporal events in a way that the correlation in the events' observations and and time is taken into account. Introducing this correlation as part of the model appears to be the main strength of the proposed approach. In short, both arrival time correlation and observation correlation are modelled. In details, the authors use HMM to model the correlation between the events' observations and the Hawkes process to model the non-fixed inter-arrival times and correlation in events' arrival times. As a second contribution of the paper at hand, the authors provide an inference algorith

Qualitative Assessment

I think the paper is promising and with a good experimental section. However, the authors need to address some points. My main concern with this work is that it is not well presented/explained. The tools used are not explained in fashion that helps the reader understand what they model in the problem they are trying to solve. For instance, the HDP in Section 2.2, for $\pi_k$ should be clearly stated that $k$ runs to (countably) infinity and indicates the number of possible states. The same applies on the motivation behind the experiments. For instance in section 5.2 it would be nice to state what the events are (on and off ?).

Confidence in this Review

2-Confident (read it all; understood it all reasonably well)


Reviewer 3

Summary

The authors propose an infinite hidden semi-Markov modulated interaction point process. This extends the infinite HMM and Hawkes processes by all assigning each state s an interaction kernel psi_s. The rate of the next state transition is the sum of the kernels of all earlier events. Having sampled time of the state transition, the new state is sampled from the corresponding row of the (infinite) state transition matrix.

Qualitative Assessment

I'm not sure if the proposed model can really be described as an "Infinite hidden semi-Markov modulated interaction point process". It is not a point process whose intensity is modulated by a semi-HMM, rather it is a marked Hawkes process whose marks follow a Markov chain (rather than being independent). Given how complex inference is for this model, some assessment of MCMC convergence would be helpful in the experiments. E.g. how long does it take? Synthetic experiment: How is the HDP-HMM used to model continuous-time data when it is a discrete-time model? For such a complex model, identifiability is an issue, so it's useful to mention How many observations were generated. Also, how long did MCMC take? Maybe it's also useful to plot CPU time as the x-axis for figure 2(1). Hamming distance: I don't really understand this, please elaborate what this means in the context of the present task. In particular, does measuring Hamming distance in the real world experiments require access to some ground truth? While the model outperforms competitors, there is a concern about overfitting. It might be useful to report some qualitative aspects of the learned quantities. E.g. what sort of kernels does the model learn for washing machines vs dryers? How does an infinite semiHMM that gives all states the same kernel perform? It would be nice to understand what aspect of the model are important for the modeling tasks. Eq 1: do we need lambda? If so, define it The variable sequence pi_k indicates the latent state sequence: should be s_n Line 139 "are a" Eq 4: Calling the waiting time PP is non-standard and not really accurate line 235: for notation simplicity line 371: intension

Confidence in this Review

2-Confident (read it all; understood it all reasonably well)


Reviewer 4

Summary

This paper develops a joint model for the occurrance times and observed outcomes of discrete events: for example the onset time and electricity usage for variance appliances or the posting time and content of messages on a social network. The model is a sensible extension of a hidden Markov model (HMM) where each state has the standard transition and emission probabilities as well as a state-specific Hawkes process which generates the time until the next event. Using a hierarchical DP prior, the number of active states can be inferred from data. To fit the proposed model to data, a particle Gibbs sampler algorithm is developed. The observed data in a sequence includes a time value t_n and an observed vector y_n at each discrete index n. The sampler must infer the state sequence s_1 ... s_N as well as the emission parameters, transition parameters, and Hawkes process parameters. Particle methods are used to sample the state sequence given all other variables, and then the remaining variables have standard Gibbs updates, except for the triggering kernels \psi and background rate \mu of the Hawkes process, which use random-walk Metropolis proposals. Experiments compare the proposed method, called iHSMM-IPP, agains several baselines from the literature including the sticky HDP-HMM (Fox et al) and the semi-Markov HDP-HSMM (Johnson & Willsky '14) and two methods (variational [11] and Metropolis sampling [17]) for the marked Hawkes process. On toy data generated from the proposed model, the proposed algorithm recovers noticeably better Hamming distance scores. On an energy disaggregation task (REDD) and a pipe burst event task, the proposed method does better as well, likely due to its joint modeling of both event times and event emissions.

Qualitative Assessment

Overall I recommend accepting this paper as a poster presentation. I can see NIPS attendees benefiting from learning more about the model, inference algorithm, and the practical applications to event failure modeling. My chief concerns are questions about scalability and lack of within-method experiments to validate different features of the algorithm. The technical content could also use revision to improve readability, though the paper is organized well. Novelty and related work ======================== This is the first paper I know of that combines Hawkes processes, HMMs with semi-Markov durations, and nonparametric priors like the HDP. However, it does look like there is some missing citations to previous efforts to synthesize Hawkes processes and state space models. See for example this somewhat paper on a switching state model where each state has a Hawkes process, which might be worth mentioning in related work (I haven't read carefully): Markov-modulated Hawkes process with stepwise decay Wang, Bebbington, and Hart Ann Inst Stat Math (2012) 64:521–544 http://www.ism.ac.jp/editsec/aism/pdf/10463_2010_Article_320.pdf On the inference side, this looks like a nice application of particle Gibbs methods. I hope the authors can share code so others can take advantage of this contribution. The description in the paper itself is likely too dense to easily allow reproducing this algorithm. Algorithm feedback ================== Sec. 4 provides lots of gory detail about the various substeps of the algorithm, but is somewhat lacking in high-level description. In particular, I'd like to see discussion of the overall runtime complexity of the sampler: What is the big-oh cost of the particle sampling of the state sequence? How does it scale with sequence length, number of states K, number of particles, etc? How would it compare with standard forward-backward type inference for simpler HMM models? How does the implementation do in a practical sense? That is, can you provide details like: it takes ___ seconds to handle a sequence of length ___ with ___ states. What kind of hyperparameters/settings is the algorithm sensitive to? I'd expect that the number of particles needs to be set somewhat carefully to balance speed and accuracy. How is this chosen? (I didnt see it anywhere, apologies if I missed it). How exactly is the algorithm initialized? I'd expect performance to be quite sensitive to initialization, especially if the number of active states is much lower than it needs to be. The details of the truncated ancestor resampling (Sec 4.2) were confusing to me. What is being truncated? Why is this necessary? Big picture details would be useful here. Experiment feedback =================== Overall I was pleased with the experiments: they include comparison to a large set of reasonable baselines and consider two challenging datasets (Redd and Pipe). I thought the biggest weakness was a lack of comparison to different possible settings of the proposed method: for example, using different numbers of particles, or with or without some hyperparameter resampling, etc. There are many speed/accuracy tradeoffs that could be made in practice, and the present paper doesn't provide much guidance about how to manage these concerns, or even indicate that such concerns should exist. For the Pipe data experiments that predict the last 10% of each sequence given the first 90%, I am confused how the number of failure events is estimated for methods like the HDP-HMM that don't have any built-in model for durations. Can you clarify? For REDD data, it would be interesting to visualize and explore qualitative relationships discovered by the learned triggering kernels. For example, is there a clear relationship between washing machine and dryer events, as expected? Question: for REDD data, are the Hamming and LogLik score reported on the training set, or a heldout set? Please clarify. Seems that these numbers are for the training set of the Pipe data. I'm curious how much the poor performance of other methods might be due to local optima, instead of modeling issues. To test this, I'd suggest initializing the HDP-HMM sampler or some other baseline method from the final state of the proposed iHSMM-IPP, or even from the "ground truth" labels, and report the Hamming distance after the sampler has run for a while. Presentation feedback ==================== I like Fig 1 (illustrating model features alongside the graphical model) a lot. Thanks! Line 196 typo: "infinite countable with the HDP" should be countably infinite The graphical model in Fig 1 uses symbol T to indicate the total length of the sequence, but later the symbol N is used. Would be good to just pick one. Fig 2: Plot on far left is missing x-axis ticks (# iterations). Please fix.

Confidence in this Review

2-Confident (read it all; understood it all reasonably well)


Reviewer 5

Summary

The paper proposes a Bayesian nonparametric model for modeling temporal events. The model uses HDP for modeling the discrete hidden states; the observation emission and the triggering mechanism are then determined conditions on these hidden states. The time of the events follows an interaction point process model. The authors propose a PMCMC inference scheme for the model since MCMC and SMC do not work reliably in their setting with highly correlated variables. They compare the performance of their model to some reasonable baselines and demonstrate the advantages of their model.

Qualitative Assessment

Technical quality: The model introduced in the paper is extensively compared to other alternatives and is shown to be quite advantageous in terms of Hamming distance and loglikelihood. My only minor comment would be to compare to (or at least mention in the related work) some continuous time processes that model both the states and the event times. Models such as Markov Modulated Poisson processes or Markov Jump Processes (and their nonparameteric variants such as models introduced in Stimberg et al. NIPS 2014 and Saeedi & Bouchard NIPS 2011) can be examples of those baselines. Novelty/originality: Although modeling both the hidden states and event times in continuous time setting is not new for Bayesian nonparametric modeling; I believe modeling the event times as IPP and conditioning that on hidden states is novel and useful. Potential impact or usefulness: The model is useful in many practical settings (as the authors have shown in their experiments) where a set of events can trigger another set of events. One other potential application of their model is analyzing neural data where such a behavior is expected from neurons. Clarity and presentation: The paper is well written and clearly explained. I only have two minor comments: 1) It's not clear to me, how the authors have used the HDP-HSMM baseline for the experiment in 5.3: what is the duration distribution? Have you discretized the time to use HDP-HSMM or you have just ignored the event times? 2) How the hyperparemters are set for the model and the baselines in the experiments? Particularly for the HDP-HSMM baseline the duration distribution and it's hyperparameters can affect the performance of the model significantly.

Confidence in this Review

2-Confident (read it all; understood it all reasonably well)